# Usefulness of Cochrane Reviews in Clinical Guideline Development—A Survey of 585 Recommendations

**DOI:** 10.3390/ijerph19020685

**Published:** 2022-01-07

**Authors:** Christoffer Bruun Korfitsen, Marie-Louise Kirkegaard Mikkelsen, Anja Ussing, Karen Christina Walker, Jeanett Friis Rohde, Henning Keinke Andersen, Simon Tarp, Mina Nicole Händel

**Affiliations:** 1The Danish Health Authority, Islands Brygge 67, 2300 Copenhagen, Denmark; mkmk@sst.dk (M.-L.K.M.); anju@sst.dk (A.U.); karen.christina.walker@regionh.dk (K.C.W.); jeanett.friis.rohde@regionh.dk (J.F.R.); hkan@sst.dk (H.K.A.); sita@SST.DK (S.T.); mina.nicole.holmgaard.handel@regionh.dk (M.N.H.); 2The Parker Institute, Bispebjerg and Frederiksberg Hospital, 2000 Frederiksberg, Denmark

**Keywords:** Cochrane, systematic review, GRADE, clinical practice guideline, core outcome set

## Abstract

The Danish Health Authority develops clinical practice guidelines to support clinical decision-making based on the Grading of Recommendations Assessment, Development, and Evaluation (GRADE) system and prioritizes using Cochrane reviews. The objective of this study was to explore the usefulness of Cochrane reviews as a source of evidence in the development of clinical recommendations. Evidence-based recommendations in guidelines published by the Danish Health Authority between 2014 and 2021 were reviewed. For each recommendation, it was noted if and how Cochrane reviews were utilized. In total, 374 evidence-based recommendations and 211 expert consensus recommendations were published between 2014 and 2021. Of the 374 evidence-based recommendations, 106 included evidence from Cochrane reviews. In 28 recommendations, all critical and important outcomes included evidence from Cochrane reviews. In 36 recommendations, a minimum of all critical outcomes included evidence from Cochrane reviews, but not all important outcomes. In 33 recommendations, some but not all critical outcomes included evidence from Cochrane reviews. Finally, in nine recommendations, some of the important outcomes included evidence from Cochrane reviews. In almost one-third of the evidence-based recommendations, Cochrane reviews were used to inform clinical recommendations. This evaluation should inform future evaluations of Cochrane review uptake in clinical practice guidelines concerning outcomes important for clinical decision-making.

## 1. Introduction

Clinical practice guidelines (hereafter referred to as guidelines) are needed to assist clinicians and policymakers in making informed decisions on healthcare and public health policy based on current best evidence [1,2,3,4]. High-quality guidelines can support medical decision making and improve care by identifying practices that maximize benefit and minimize harm [5]. High-quality guidelines are developed with methodological rigor by multidisciplinary panels. Guidelines consider several factors such as patients’ preferences and values, available resources and costs, and cultural heterogeneity, in addition to the evidence [6].

Conducting or using existing high-quality systematic reviews to synthesize current best evidence is essential in guideline development [4]. Cochrane reviews are regarded as high-quality summaries presenting unbiased information useful for developing guidelines and, thus, impact clinical decision-making [7,8].

Cochrane reviews are highly warranted in guideline development and can save guideline developers time (e.g., using existing searches/data). By following high methodological standards [9,10,11,12], the Cochrane Review Groups have published more than 8704 systematic reviews across all health-related topics [13], and Cochrane reviews have informed both national and international guidelines [14,15,16].

However, utilizing Cochrane reviews in guideline development is challenging if there are large discrepancies between the scope of the review and the guideline [17]. Discrepancies are often found in the population of interest, intervention, comparison and/or outcomes of interest. For the findings to influence policy and practice, the outcomes need to be especially relevant and important to key stakeholders, including patients and the public, health care professionals and others making decisions about health care [18,19,20,21]. Examples of critical outcomes not reported as standard in clinical trials and Cochrane reviews have been reported [22,23]. Standardization of critical and important outcomes is largely available in Core Outcome Set (COS), an agreed standardized set of outcomes that should be measured and reported, as a minimum, in all clinical trials in specific health areas [18,24]. However, the uptake of COS in both Cochrane reviews [23,25] and clinical trials is are not yet consistent [26,27].

To the best of our knowledge, no guideline development group has investigated how Cochrane reviews have informed specific critical and important outcomes of interest in guidelines. Investigation of the extent to which Cochrane reviews are used in guidelines has previously been done by citation searches [15,16]. However, this method can affect the validity of the findings since Cochrane reviews can inform both the background of the guideline and specifically inform the systematic searches for relevant evidence [28]. Thus, the objective of this study was to explore the usefulness of Cochrane reviews as a source of evidence in the development of clinical recommendations.

The methodology used in the Danish Health Authority guideline development has previously been described [29,30,31,32,33,34,35,36,37,38,39,40]. In short, the methodology is based on the recommendations for conducting systematic reviews and meta-analyses from the Cochrane Handbook for systematic reviews [8]. Since 2013, the Grading of Recommendations Assessment, Development, and Evaluation (GRADE) methodology has been used [41]. GRADE has become the most recognized and transparent method available to date to develop clinical guidelines. The clinical questions are operationalized according to the population, intervention, comparator, outcomes format (PICO), and pre-specified explicitly in a protocol approved by the management of the Danish Health Authority. Outcomes are judged as critical or important to patients, and their timing and effect measures are defined a priori [19,41]. According to GRADE, a critical outcome is defined as being patient-important and critical for decision making. The overall quality of the evidence depends on the certainty of the evidence of the critical outcome(s). An important outcome is defined as important to patients but not critical for decision making [19]. Outcomes are preferably chosen from COS [24,42,43,44,45,46] if available, or the guideline panel will agree on critical and important outcomes based on outcomes used in clinical practice and clinical trials. Outcomes are always prioritized and adapted to national contextual factors [19]. The Danish national recommendations are continuously monitored; at least every three years, an updated guideline search and search for Cochrane reviews will inform a need to update the recommendation. The methodology for conducting the systematic reviews within the guidelines process is the same used for the entire study period.

## 2. Materials and Methods

Guidelines published by the Danish Health Authority between 2014 and 2021 were identified and reviewed [47]. The total number of recommendations included in the Danish guidelines were mapped and categorized into evidence-based recommendations or expert consensus recommendations. The full guidelines are available in Danish with all supporting material on the Danish Health Authority website [29] and the MAGICapp website [48].

Data from the recommendations were extracted to a pilot-tested Microsoft Excel datasheet (see Appendix A). The pilot-testing included data checking by one reviewer (C.B.K.) of the first five guidelines with extracted data (starting from 2014). Any discrepancies resulted in continuous adjustments to the datasheet. One out of four reviewers (C.B.K., M.L.K.M., A.U., J.F.R.) followed a series of steps to extract information about how recommendations were based on evidence derived from Cochrane reviews. First, we identified the Cochrane review citations by screening the reference list of the guidelines. Secondly, we noted whether the use of the Cochrane review in the guideline was applied either as background information or included as a source of evidence. Finally, we extracted information from the results section and assorted the outcomes into the following four non-overlapping categories: (I) all outcomes (both critical and important outcomes) informed by a Cochrane review, (II) all critical outcomes informed, but not all important outcomes (III) at least one critical outcome informed, or (IV) no critical but one or more important outcomes informed by a Cochrane review. Each recommendation was only categorized once.

The distributions and proportions of recommendations informed by Cochrane reviews across topics were summarized with median and interquartile range (IQR). We analyzed whether the proportion of evidence-based recommendations in the Danish guidelines varied over time. Moreover, the proportions of recommendations with evidence from Cochrane reviews across medical topics (using the Cochrane Topic taxonomy [49]) were also analyzed. The analyses were performed in Microsoft Excel (Microsoft Cooperation, Redmond, United States, 2019).

## 3. Results

Between 2014 and 2021, 56 guidelines were published by the Danish Health Authority, comprising 374 (64%) evidence-based recommendations and 211 (36%) expert consensus recommendations (Table 1). The 374 recommendations included 107 citations of Cochrane reviews, where 106 (28%) were included as evidence (Table 1). Cochrane reviews informed all outcomes in 28 (7%) evidence-based recommendations (Table 1). For 36 (10%) of the evidence-based recommendations, all critical outcomes were informed by a Cochrane review, but not all of the important outcomes were. In 33 (9%) of the evidence-based recommendations, at least one critical outcome was informed by Cochrane reviews. Finally, nine (2%) evidence-based recommendations had at least one important outcome informed by evidence from a Cochrane review.

When a Cochrane review was available and used as a source of evidence, the Cochrane review informed all critical outcomes and important outcomes in 26% of all recommendations informed by Cochrane reviews (Table 2). A total of 34% informed all critical but not all important outcomes, 31% informed at least one critical outcome and 8% informed at least one important but no critical outcome (Table 2).

From 2014 to 2021, the proportion of evidence-based recommendations in the Danish guidelines varied from 31% in 2014 to 81% in 2021 (Figure 1). The proportion of recommendations with evidence informed by Cochrane reviews varied from 24% in 2014 to 20% in 2021 (range 17–42%) (Figure 1). Recommendations with evidence from Cochrane reviews varied between 0 and 80% across medical topics, with a median of 29% (IQR: 14–38%) (Figure 2). Across topics, Pregnancy and Childbirth (75%), Pain and Anesthesia (63%) and Dentistry and Oral Health (40%) had the highest use of evidence from Cochrane reviews. In contrast, the topics Eyes and Vision (11%), Endocrine and Metabolic (5%) and Skin Disorders (0%) had the lowest.

## 4. Discussion

Since 2014, recommendations published by the Danish Health Authority have increasingly been based on evidence rather than consensus on best practice. However, less than one-third of the evidence-based recommendations published by the Danish Health Authority used evidence from Cochrane reviews to inform outcomes in the clinical question of interest. The use varied across medical topics. In 17% of these recommendations, evidence from Cochrane reviews was used for all critical outcomes. However, in 93% of the evidence-based recommendations, the Cochrane reviews did not inform all outcomes. However, when a Cochrane review was used as evidence in a recommendation, in 60% of these recommendations it informed all critical outcomes (when merging the two categories: “all critical and important outcomes covered” and “all critical outcomes covered/informed”). Additionally, in 91%, at least one critical outcome was informed (when merging all categories except “at least one important but no critical outcome”).

There are several possible explanations for the sparse use of Cochrane reviews in the evidence-based recommendations reviewed. One explanation could be the lack of updated Cochrane reviews [15]. The Cochrane collaboration aims to impact global health by publishing updated systematic reviews. Thus, the Danish National Health Authority expects Cochrane reviews to not only be performed to the highest standards but also include all patient-relevant outcomes. However, Cochrane’s relevance and value will depend on its ability to respond innovatively and promptly to decision-makers’ needs. This is confirmed in experiences reported from guideline development groups from the United Kingdom National Institute for Health Research [15,17].

Another explanation for the findings in our study might be related to the aim of the Danish guidelines, such as specific eligibility criteria related to population and/or intervention [15]. Another aspect could be that nationally developed guidelines often adapt clinical questions to the context of the healthcare system in that country. For example, the Danish Health Authority ensures that clinical questions in the guidelines cover the following criteria: considerable health- or resource-burden, variation in clinical practice, professional disagreement or doubt about best practice, new precarious technology, or considerable possible change in treatment/screening indication. These criteria ensure that clinical questions are relevant for clinical practice [50]. Therefore, the recommendations deal with selected well-defined aspects of diagnostics, treatment, care and rehabilitation for specific patient groups, where the need for clarifying the evidence has shown to be relevant. It is emphasized that the topics address important issues, i.e., either controversial, unresolved, or intrusive, and not trivial or already well-described issues.

A major threat to the validity of clinical trials and systematic reviews is missing outcome data or inconsistent reporting of outcome data in clinical research [51,52,53,54]. In one-fifth of 283 Cochrane reviews, more than 50% of the patient data for the primary outcome was missing [55]. Cochrane reviews and guidelines often include many outcomes about the harms and benefits of an intervention or management strategy [19,56,57]. When choosing to include a Cochrane Review as a source of evidence, it is enough if the review has just one relevant outcome of interest. Cochrane states that established COS should be used where available, and patient-reported outcomes should be included where possible [7,56]. However, evidence suggests that these guidelines are not consistently followed [23,25,58].

This study showed a tendency to include fewer Cochrane reviews in evidence-based recommendations between 2014 and 2021. This was surprising since guideline developers following the GRADE method are likely more prone to include Cochrane reviews [16]. This may be due to the changed criteria or priority of clinical questions for the guideline development at the Danish Health Authority since the GRADE method was incorporated in 2013 [50]. Here, the clinical questions selected were the ones that are difficult to answer in standard clinical trial designs, thus hypothetically leading to less available evidence to answer the question and possibly limiting the transferability to a Cochrane review.

A limitation to this study was that only one author extracted information per guideline, and thus there was no validation of the extracted information. Additionally, the authors did not extract information about the uptake of COS in the Danish guidelines, and this information could have informed the discussion of the findings. It is worth noting that there has not been a standardized use nor citation of COS across guidelines in the development of the Danish guidelines in this period.

This descriptive study focused on outcomes informed by evidence from Cochrane reviews. More investigation of the use of systematic reviews (not limited to Cochrane reviews) and specific use of COS could have further clarified the findings and informed future evaluations. Unfortunately, information on the use of COS was only reported in a few Danish guidelines, primarily the latest published.

The descriptive nature of this study calls for further investigation of how Cochrane reviews are used to inform outcomes of interest in guidelines in other countries using GRADE. Additionally, few studies have been undertaken to assess the uptake of COS in trials and reviews, and to our knowledge, no studies have assessed the uptake of COS in guidelines [26]. This could help facilitate international collaborations.

Rather than simply presenting the status of applying Cochrane reviews (the objective of this study), the next step might be investigating the effect of using Cochrane reviews on methodology process parameters, such as time and resources. Such evaluation could be relevant when investigating adaption of Cochrane reviews (using both evidence-base, risk of bias assessments and analyses). Adapting Cochrane reviews or other high-quality systematic reviews is highly required since this, besides saving guideline developers resources, can also help limit research waste [59,60,61]. However, a barrier to adapting reviews is the lack of transparency on data extraction, i.e., clarity on which study data were applied in meta-analyses when multiple data are available for the same outcome. This often leads guideline developers to not favor review adaption [17]. Standardized methods for data extraction [62] and prioritizing specific outcome measures is warranted, e.g., when multiple patient-reported outcome measures within a domain are available [63]. Future studies should investigate the uptake of COS in trials, reviews and guidelines, and the use of adaptions in guideline development.

## 5. Conclusions

Almost one-third of the Danish Health Authority evidence-based recommendations used Cochrane reviews to inform clinical recommendations. For nearly two-thirds of these recommendations, Cochrane reviews were used for all critical outcomes. This emphasizes the demand for using core outcome sets in clinical trials, Cochrane reviews and guidelines. This evaluation should inform future evaluations of Cochrane review uptake in clinical practice guidelines concerning outcomes critical and important for clinical decision-making.

## Figures and Tables

**Figure 1 ijerph-19-00685-f001:**
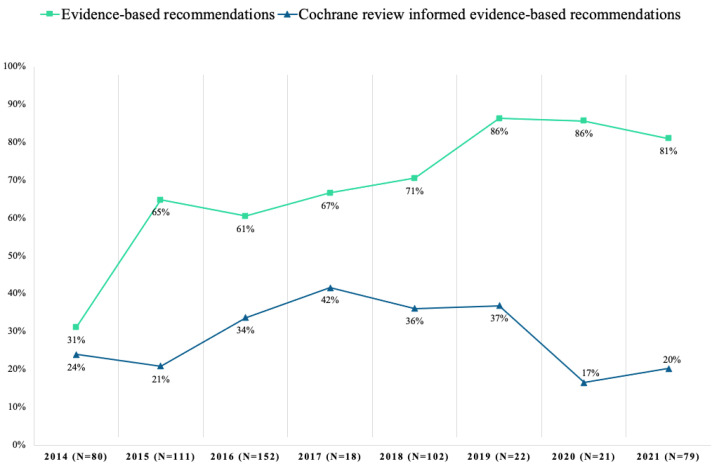
Evidence-based recommendations with outcomes informed by evidence from Cochrane reviews in Danish National Clinical Guideline development from 2014–2021.

**Figure 2 ijerph-19-00685-f002:**
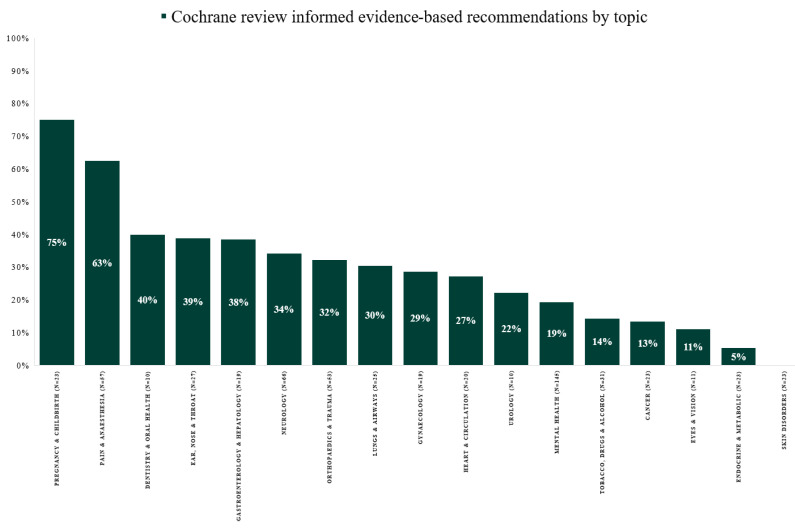
Evidence-based recommendations informed by evidence from Cochrane reviews by topic in Danish National Clinical Guideline development from 2014 to 2021.

**Table 1 ijerph-19-00685-t001:** Recommendations in Danish National Clinical Guideline development between 2014 and 2021.

		Total Recommendations	Total Evidence-Based Recommendations
	n	585	374
Total guidelines	56		
Total recommendations	585		
Evidence-based recommendations	374	64%	
Best practice ^1^ recommendations	211	36%	0%
Recommendations informed by Cochrane review	106	18%	28%
Categories of outcomes			
(I) All critical and important outcomes covered ^2^	28	5%	7%
(II) All critical outcomes covered/informed ^2^	36	6%	10%
(III) At least one critical outcome covered/informed ^2^	33	6%	9%
(IV) At least one important but no critical outcome ^2^	9	2%	2%

^1^ Recommendation by expert consensus only. ^2^ Non-overlapping categories.

**Table 2 ijerph-19-00685-t002:** Proportions of categories of outcomes informed by Cochrane reviews in Danish National Clinical Guideline development.

		Recommendations Informed by Cochrane Review
Categories of Outcomes	n	106
(I) All critical and important outcomes covered ^1^	28	26%
(II) All critical outcomes covered/informed ^1^	36	34%
(III) At least one critical outcome covered/informed ^1^	33	31%
(IV) At least one important but no critical outcome ^1^	9	8%

^1^ Non-overlapping categories.

## Data Availability

The data presented in this study are available in Appendix A.

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
