# Peer review of "Usefulness of Cochrane Reviews in Clinical Guideline Development—A Survey of 585 Recommendations"

_ijerph, 2022, doi:10.3390/ijerph19020685_

Round 1
Reviewer 1 Report
I think this research is an interesting topic with regards to the utilization of Cochrane reviews. However, compared to the results of analyzing a large amount of data, the meaningful results presented to the readers seem to be very weak. It seems that the analysis results are too superficial, it does not provide academic or in-depth content.
o Materials and Methods
You mention that “finally, we screened the summary of findings table and narrative evidence section for information about how each outcome was informed by evidence from a Cochrane review.” Please describe this in detail.
Please describe in detail what the researchers suggests in the research materials and method in the research result.
Rather than simply presenting the current status of applying Cochrane reviews, I think that it will be a more academic paper to statistically compare the effectiveness of the applied group and the non-applied group of the Cochrane reviews.
o Results
The frequencies and percentages presented in Table 1 are very confusing. Please indicate the frequency and percentage in all fields, and correct to match the frequency and percentage each other.
It is questionable whether the contents specified in the research method of this study are sufficiently presented in this research results.
o References
Please add DOI to your references.
Author Response
1. I think this research is an interesting topic with regards to the utilization of Cochrane reviews. However, compared to the results of analyzing a large amount of data, the meaningful results presented to the readers seem to be very weak. It seems that the analysis results are too superficial, it does not provide academic or in-depth content.
Author’ reply: We want to thank the reviewer for the effort of reviewing this manuscript. We appreciate the reviewer’s interest in the topic of our paper, and we have tried to respond to the specific comments.
We acknowledge that the study results are descriptive in nature, but this agrees with the objective. We hope that this study will promote more investigation into the prevalence of Cochrane review uptake in guidelines, but more importantly, the influence of using core outcome sets in both Cochrane reviews and guideline development.
Action Taken: We have made all data available in the supplementary material to facilitate our readers’ wider use of our data.
2. Materials and Methods. You mention that “finally, we screened the summary of findings table and narrative evidence section for information about how each
outcome was informed by evidence from a Cochrane review.” Please describe this in detail.
Author’ reply: Thank you for this comment. We agree that this has not been sufficiently described. Therefore, we have elaborated and profoundly revised the methods section to describe the data extraction method in more detail.
Action Taken: Revised section in the materials and methods section (page 3,
line no. 100-114): Data from the recommendations were extracted to a pilottested Microsoft Excel datasheet (see supplementary appendix
1). The pilot-testing included data checking by one reviewer (C.B.K.) of the first five guidelines with extracted data (starting from 2014). Any discrepancies resulted in continuous adjustments to the datasheet. One out of four reviewers
(C.B.K., M.L.K.M., A.U., J.F.R.) followed a series of steps to extract information about how recommendations were based on evidence derived from Cochrane reviews. First, we identified the Cochrane review citations by screening the reference list of the guidelines. Secondly, we noted whether the use of the
Cochrane review in the guideline was applied either as background information or included as a source of evidence.
Finally, we extracted information from the result section for information on the following four non-overlapping categories of outcomes: I) all outcomes (both critical and important outcomes) informed by a Cochrane review, II) all critical
outcomes in-formed, but not all important outcomes III) at least one critical outcome informed, or IV) no critical but one or more important outcomes informed by a Cochrane review. Each recommendation was only categorized once.
3. Please describe in detail what the researchers suggests in the research materials and method in the research result.
Author’ reply: We acknowledge that the description of the research materials and methods were not sufficiently linked to the research results in the original draft. We have profoundly revised the materials and methods section substantially to improve the agreement between the two sections of the manuscript.
Action Taken: See revised materials and methods section (page 2-3, line no.94-120) and results section (page 3, line no. 122-146)
4. Rather than simply presenting the current status of applying Cochrane reviews, I think that it will be a more academic paper to statistically compare the
effectiveness of the applied group and the non-applied group of the Cochrane reviews.
Author’ reply: Thank you for this comment and suggestion – a very relevant point.
The objective of this study was to explore the usefulness of Cochrane reviews as a source of evidence in the development of clinical recommendations. Although it would be highly interesting, unfortunately, we do not have monitoring data
available. However, we are engaged to improve the implementation and monitoring of guidelines useability. We have included a section about future investigations in the discussion section.
Action Taken: Revised section in the discussion section: (page 7-8, line no.258-262): Rather than simply presenting the current status of applying Cochrane reviews (the objective of this study), the next step might be investigating the effect of using Cochrane reviews on methodology process outcomes, such as time and resources. This, however, might also be related to the adaption of Cochrane reviews (using both evidence-base, risk of bias assessments and analyses).
5. The frequencies and percentages presented in Table 1 are very confusing. Please indicate the frequency and percentage in all fields, and correct to match the frequency and percentage each other.
Author’ reply: Thank you for this comment. We have revised Table 1 accordingly and split it into two tables to improve readability.
Action Taken: See revised table 1 and table 2 (page 4).
6. It is questionable whether the contents specified in the research method of this study are sufficiently presented in this research results.
Author’ reply: Thank you for this comment. We hope that our revisions of the materials, methods, and results sections are more in line.
Action Taken: Revised section in the results section: (page 3, line no. 123-146): Between 2014 and 2021, 56 guidelines were published by the Danish Health Authority, comprising 374 (64%) evidence-based recommendations and 211 (36%) expert consensus recommendations (Table 1). The 374 recommendations included 107 citations of Cochrane reviews, where 106 (28%) were included as evidence (Table 1). Cochrane reviews informed all outcomes in 28 (7%) evidence-based recommendations (Table 1). For 36 (10%) of the evidencebased recommendations, all critical outcomes were informed by a Cochrane review, but not all important outcomes. In 33 (9%) of the evidence-based, at least one critical outcome was informed by Cochrane reviews. Finally, 9 (2%) evidence-based recommendations had at least one important outcome informed by evidence from a Cochrane review.
When a Cochrane review was available and used as a source of evidence, the Cochrane review informed all critical outcomes and important outcomes in 26% of all recommendations informed by Cochrane reviews. 34% informed all critical but not all important outcomes, 31% informed at least one critical outcome and 8% informed at least one important but no critical outcome (Table 2). From 2014 to 2021, the proportion of evidence-based recommendations in the Danish guidelines varied from 31% in 2014 to 81% in 2021 (Figure 1). The proportion of recommendations with evidence informed by Cochrane reviews varied from 24% in 2014 to 20% in 2021 (range 17–42%) (Figure 1). Recommendations with evidence from Cochrane reviews varied between 0-80% across medical topics, with a median of 29% (IQR: 14–38%) (Figure 2). Across topics, Pregnancy & Childbirth (75%), Pain and anaesthesia (63%) and Dentistry and oral health (40%) had the highest use of evidence from Cochrane reviews. In contrast, Eyes and vision (11%), Endocrine and metabolic (5%) and Skin disorders (0%)
had the lowest.
7. Please add DOI to your references.
Author’ reply: Thank you for highlighting this. We have added DOI to the references.
Action Taken: DOI information has been added to references

Reviewer 2 Report
Dear authors,
The study investigated the usefulness of Cochrane reviews as a source of evidence in developing clinical recommendations. However, using Cochrane reviews in guideline development is difficult. For outcomes to influence policy and practice, especially externally, they need to be relevant and important to key stakeholders including patients and audiences, healthcare professionals and others
I have a few questions
Why the authors considered the review from 2014, health recommendations have changed over the years and some recommendations are far out of date, so they should not be taken into account. - explain that
Can you explain why 60% of the Cochrane reviews had critical results and only 8% did not report any critical results. Please refer to this in more detail.
I do not see standardized methods of data extraction, the methodology is not very clear to me. Did the authors use any standardized method?
Author Response
1. The study investigated the usefulness of Cochrane reviews as a source of evidence in developing clinical recommendations. However, using Cochrane reviews in guideline development is difficult. For outcomes to influence policy and practice, especially externally, they need to be relevant and important to key
stakeholders including patients and audiences, healthcare professionals and others. I have a few questions.
Author’ reply: We want to thank the reviewer for the effort of reviewing this manuscript.
Action Taken: We do agree that the present study highlights an important area.
We have included a comment in the discussion about the expectations from The Danish National Health Authority for the quality of Cochrane Reviews.
Revised section in the discussion section (page 6, line no.200-203): The Cochrane collaboration aims to impact global health by publishing updated systematic reviews. Thus, the Danish National Health Authority expects Cochrane reviews to not only be performed to the highest standards but also include all
patient-relevant outcomes.
2. Why the authors considered the review from 2014, health recommendations have changed over the years and some recommendations are far out of date, so
they should not be taken into account. - explain that.
Author’ reply: Thank you for this comment. We have described the methodology used in the period (2014-2021) in the guideline development (line no. 73-93). The reason for selecting this study period was that the methodology in the Danish guideline setting was the same. The Danish guideline recommendations are continuously monitored for update every three years, which, as a minimum, include a search for other guidelines and Cochrane reviews. We have included a section about this in the introduction.
Action Taken: Revised section in the introduction section (page 2, line no.73-93): The methodology used in the Danish Health Authority guideline development has previously been described 29–40. In short, the methodology is based on the recommendations for conducting systematic reviews and meta-analysis from
Cochrane Handbook for systematic reviews 8. Since 2013, the Grading of Recommendations Assessment, Development, and Evaluation (GRADE) methodology has been used 41. GRADE has become the most recognized and transparent method available to date to develop clinical guidelines. The clinical
questions are operationalized according to the population, intervention, comparator, outcomes format (PICO), and pre-specified in an explicit protocol approved by the management of the Danish Health Authority. Outcomes are judged as critical or important to patients, and their timing and effect measures
are de-fined a priori 19,41(p1). According to GRADE, a critical outcome is defined as being patient-important and critical for decision making. The overall quality of the evidence depends on the certainty of the evidence of the critical outcome(s). An important outcome is defined as important to patients but not
critical for the decision making 19. Outcomes are preferably chosen from COS 24,42–46 if available, or the guideline panel will agree on critical and important outcomes based on outcomes used in, i.e. clinical practice and clinical trials.
Outcomes are always prioritized and adapted to national contextual factors 19. Recommendations are continuously monitored; at least every three years, an updated guideline search and search for Cochrane reviews will inform a need to
update the recommendation. The methodology for conducting the systematic reviews within the guidelines process is the same used for the entire study period.
3. Can you explain why 60% of the Cochrane reviews had critical results and only 8% did not report any critical results. Please refer to this in more detail.
Author’ reply: We acknowledge this comment. When a Cochrane review was used, 60% informed all critical outcomes (when collapsing the two categories: “all critical and important outcomes covered” and “all critical outcomes covered/informed”). Also, in 91%, at least one critical outcome was informed (when collapsing all categories except “at least one important but no critical outcome”).
Action Taken: We have revised the results and discussion section and ultimately presented the data in more detail.
Revised section in the discussion section (page 6, line no.186-197):
However, when a Cochrane review was used, 60% informed all critical outcomes (when collapsing the two categories: “all critical and important outcomes covered” and “all critical outcomes covered/informed”). Also, in 91%, at least one
critical outcome was informed (when collapsing all categories except “at least one important but no critical outcome”).
4. I do not see standardized methods of data extraction, the methodology is not very clear to me. Did the authors use any standardized method?
Author’ reply: Thank you for this comment, which is in line with other comments from the reviewers. We have revised the methods section to describe the data extraction method in more detail.
As described in the discussion section (limitations), we did not use a standardized method.
Please see review 1 response 2 for related review response.
Action Taken: Revised section in the materials and methods section (page 3, line no. 100-114): Data from the recommendations were extracted to a pilottested Microsoft Excel datasheet (see supplementary appendix 1). The pilot-testing included data checking by one reviewer (C.B.K.) of the first five guidelines with extracted data (starting from 2014). Any discrepancies resulted in continuous adjustments to the datasheet. One out of four reviewers (C.B.K., M.L.K.M., A.U., J.F.R.) followed a series of steps to extract information about how recommendations were based on evidence derived from Cochrane reviews. First, we identified the Cochrane review citations by screening the reference list of the guidelines. Secondly, we noted whether the use of the Cochrane review in the guideline was applied either as background information or included as a source of evidence.
Finally, we extracted information from the result section for information on the following four non-overlapping categories of outcomes: I) all outcomes (both critical and important outcomes) informed by a Cochrane review, II) all critical
outcomes in-formed, but not all important outcomes III) at least one critical outcome informed, or IV) no critical but one or more important outcomes informed by a Cochrane review. Each recommendation was only categorized once.

Reviewer 3 Report
Interesting article! It shows the importance of clinical experience of the experts. It seems that the meta-analysis of the Cochrane reviewers is not enough. However, the vast multitude of the recommendations can impossibly have a major impact on the clinical practice in the field of general medicine. So, the question is, to which end these guidelines are produced. Tribulation of the GPs?
Author Response
1. Interesting article! It shows the importance of clinical experience of the experts. It seems that the metaanalysis of the Cochrane reviewers is not enough.
However, the vast multitude of the recommendations can impossibly have a major impact on the clinical practice in the field of general medicine. So, the
question is, to which end these guidelines are produced. Tribulation of the GPs?
Author’ reply: We want to thank the reviewer for the effort of reviewing this manuscript, and we appreciate the reviewer’s interest in our paper.
We will try to describe the landscape for guideline development and target audience in Denmark.
In Denmark, clinical practice guidelines developed by the Health Authority differentiates from the practice guidelines developed by health professional organisations. The latter is devoted to developing treatment guidelines to assist health professionals in all parts of their work. As elaborated in this paper, the Danish Health Authority only deals with selected well-defined aspects of diagnostics, treatment, care, and rehabilitation for specific patient groups, where the need to uncover the evidence has shown to be particularly relevant. Thus, the recommendations are perceived as supplements to other existing guidelines.
The reasons for selecting a topic for a recommendation is not tribulation of GPs only, but rather in areas where there are considerable health- or resource-burden, variation in clinical practice, professional disagreement or doubt about best practice, new precarious technology, or considerable possible change in treatment/screening indication. It is emphasised that the topics address important issues, i.e., either controversial, unresolved, or intrusive, and not trivial or already well-described issues.
Although highly relevant, at this point, we do not have monitoring data available. However, we are engaged to improve the implementation and monitoring of guidelines useability prospectively in future recommendations.
We have included a section in the discussion
Action Taken: Revised section in the discussion section (page 7, line no. 214-220): These criteria ensure that clinical questions are relevant for clinical practice 50. Therefore, the recommendations deal with selected well-defined aspects of diagnostics, treatment, care and rehabilitation for specific patient groups, where the need for uncovering the evidence has shown to be particularly relevant. It is emphasized that the topics address important issues, i.e. either controversial, unresolved, or intrusive, and not trivial or already well-described issues.
Round 2
Reviewer 1 Report
You worked very hard to revise this manuscript.
Please check that there are some typos or writing format in general.
Author Response
1. You worked very hard to revise this manuscript.
Please check that there are some typos or writing format in general.
Author’ reply: We want to thank the reviewer for the effort of reviewing this
manuscript and the acknowledgement of our previous revisions.
All authors have checked the manuscript for typos and writing format, including clarity, and we hope that the revisions are in line with the reviewer’s comment.
Action Taken: See revisions throughout the manuscript.
